# Postoperative Survival and Clinical Outcomes for Uterine Leiomyosarcoma Spinal Bone Metastasis: A Case Series and Systematic Literature Review

**DOI:** 10.3390/diagnostics13010015

**Published:** 2022-12-21

**Authors:** Deyanira Contartese, Stefano Bandiera, Gianluca Giavaresi, Veronica Borsari, Cristiana Griffoni, Alessandro Gasbarrini, Milena Fini, Francesca Salamanna

**Affiliations:** 1Complex Structure of Surgical Sciences and Technologies, IRCCS Istituto Ortopedico Rizzoli, 40136 Bologna, Italy; 2Spine Surgery, IRCCS Istituto Ortopedico Rizzoli, 40136 Bologna, Italy; 3Scientific Direction, IRCCS Istituto Ortopedico Rizzoli, 40136 Bologna, Italy

**Keywords:** bone metastasis, spine, uterine leiomyosarcoma, clinical–pathological characteristics, survival

## Abstract

Spinal bone metastases from uterine leiomyosarcoma (LMS) are relatively uncommon and few data are present in the literature. In this study, cases of nine consecutive patients who underwent spinal surgery for metastatic uterine LMS between 2012 and 2022 at a single institution were retrospectively reviewed. The recorded demographic, operative, and postoperative factors were reviewed, and the functional outcomes were determined by changes in Frankel grade classification during follow-up. A systematic review of the literature was also performed to evaluate operative and postoperative factors and outcomes for patients with the same gynecological metastases to the spine. For our cases, the mean time between primary tumors to bone metastases diagnosis was 5.2 years, and the thoracic vertebrae were the most affected segment. Overall, median survival after diagnosis of metastatic spine lesions was 46 months. For the systematic review, the mean time between primary tumors to bone metastases was 4.9 years, with the lumbar spine as the most involved site of metastasis. Overall, median survival after diagnosis was 102 months. Once a spinal bone lesion from LMS is identified, surgical treatment can be beneficial and successful in alleviating symptoms. Further efforts will be crucial to identify prognostic markers as well as therapeutic targets to improve survival in these patients.

## 1. Introduction

Leiomyosarcomas (LMS) are soft tissue tumors which develop mainly from smooth muscles in visceral organs, i.e., the uterus or the gastrointestinal tract, and from non-visceral structures, i.e., large- to mid-sized veins and/or dermal pilar smooth muscle in the extremities or trunk. It has an incidence of ~0.64 per 100,000 patients, with prognosis and aggressiveness varying by subtype [1]. Despite the subtypes and the high variability in the clinical course, the oncologic management of LMS is generally similar and is in the form of surgical excision since the advantages of chemotherapy and radiation are still doubtful [1,2]. LMS has a range of outcomes, mainly based on grade, with a predilection for metastasis development [3]. Based on epidemiological evidence, distant metastatic lesions represent a late manifestation of the disease, and the lungs and liver are the most common sites of LMS metastasis; however, patients with LMS also have a risk, although rare, of metastasis to bone [3]. Currently, there is a gap in the literature concerning the presence of bone metastases from LMS and their subsequent clinical courses and patients’ survival. A recent study found that bone metastases were present in 11% of patients with LMS [4]. This study also showed that the most common LMS subtypes with bone metastases were the uterine subtypes, with the axial skeleton as the most common site of bone metastasis [4]. Uterine LMS has a high prevalence in the pre- and peri-menopausal period. Total abdominal hysterectomy with or without bilateral salpingo-oophorectomy, followed by radiotherapy and/or chemotherapy, is the mainstay of the current treatment protocol for localized uterine LMS. However, several data have reported that uterine LMS is characterized by early metastasis [5,6]. The spine, most prevalently the thoracic and lumbar areas, seems to be the most common site for uterine LMS, but literature information is limited to few case reports and case series and there is not a clear consensus on treatments and on patients’ survival after treatments. Spine metastases from uterine LMS are manly osteolytic and are not easy to manage because they are highly destructive, hyper-vascularized, and resistant to chemo- and radiotherapy, thus leading to pathologic fractures and spinal cord compression that severely compromise the patient’s functional status and their quality of life [7]. The surgical options for spinal metastatic uterine LMS range from spinal decompression to en bloc excision. To obtain more accurate prognostic information on uterine LMS spinal bone metastasis, we retrospectively reviewed the medical records of patients who had undergone surgery for spinal metastases of uterine LMS in the last ten years at our institution and performed a systematic literature review to identify all published reports on such lesions.

## 2. Materials and Methods

### 2.1. Case Series

The present study is a retrospective case series completed from data available from the Spine Surgery Unit of IRCCS Istituto Ortopedico Rizzoli and approved by the Ethics Committee of the Emilia Romagna Region (Comitato Etico Indipendente Area Vasta Emilia Centro, CE-AVEC, Prot. n. 0007902, 07/01/2019). The study involved a total of 775 patients with vertebral bone metastases, who were surgically treated and managed from 2001 to 2021. Out of the total number, nine patients had uterine LMS spinal bone metastasis. An extensive review of medical records was performed for all of the patients to obtain specific clinical data and overall outcomes. Demographic factors, including age, race, comorbidities, smoking habits, alcohol intake, and menopausal status at diagnosis of bone metastasis were reviewed and collected. Furthermore, specific data on bone metastases, i.e., the bone metastases diagnostic tool, time from LMS primary diagnosis to bone metastasis, main bone lesion level, the presence of other bone metastasis lesions, fractures, or extra-osseous metastases, pre-operative symptoms, surgical approach and techniques, post-operative therapies, pre- and post-operative neurologic function, spinal instability, and, finally, surgical staging of bone metastasis and months until death/last follow-up after bone metastasis were reviewed and collected [8,9,10]. The pre- and post-operative and the latest follow-up neurologic function was evaluated using the Frankel grade classification, according to the degree of spinal cord injury, as follows: grade A, complete loss of sensation and motor function below the level of injury; grade B, no motor function, but some sensation is retained below the level of the lesion; grade C, some muscles below the level of injury have motor function, but no proper function is present; grade D, proper function is present below the plane of injury, walking with crutches; and grade E, typical motor and sensory function, pathological reflexes possible [8]. Spinal instability was evaluated by adopting the spinal instability neoplastic score (SINS) which helps to assess tumor-related instability of the vertebral column. It scores six variables: location of the lesion, characterization of pain, type of bony lesion, radiographic spinal alignment, degree of vertebral body destruction, and involvement of posterolateral spinal elements. The minimum score is 0, and the maximum score is 18. A score of 0 to 6 denotes stability, a score of 7 to 12 denotes indeterminate (possibly impending) instability, and a score of 13 to 18 denotes instability [10]. Finally, the surgical staging of bone metastasis was evaluated according to the Weinstein–Boriani–Biagini classification [9]. The classification describes the vertebral involvement as sections of a clock face (‘‘zones’’) centered on the spinal cord, from zone 1 (left spinous process and lamina) through zone 6 (left anterior wedge of vertebral body) and back around to zone 12 (right spinous process and lamina). In addition, the prefixes A–E are used to denote radial levels (‘‘layers’’) of vertebral involvement, from extraosseous paraspinal tissues (layer A) through to extradural (layer D) and intradural (layer E). 

### 2.2. Systematic Review

#### 2.2.1. Eligibility Criteria

The PICOS model (population, intervention, comparison, outcomes, study design) [11] was used to design this review: (1) studies that considered patients (population) with uterine LMS spinal bone metastasis submitted to, (2) spinal surgery (interventions), (3) without a comparison group (comparisons), (4) that reported post-operative clinical and/or functional outcomes and overall survival (Outcomes), in (5) all type of clinical studies (study design). All studies up to 14 July 2022 were included in this review if they met the PICOS criteria. We excluded studies that evaluated (1) primary LMS other than the uterine one (e.g., brain, spine, spinal cord), (2) patients who did not undergo spinal surgery, and (3) articles with incomplete outcomes or data. Additionally, we excluded reviews, letters, comments to editors, and articles not written in English.

#### 2.2.2. Search Strategies

Our literature review involved a systematic search conducted in July 2022. We performed our review according to the preferred reporting items for systematic reviews and meta-analyses (PRISMA) statement [12]. The search was carried out on three databases: PubMed, Scopus, and the Web of Science Core Collection. The following combination of terms was used (spine OR spinal OR vertebral) AND (bone metastasis) AND (uterine leiomyosarcoma), and for each of these terms, free words, and controlled vocabulary specific to each bibliographic database were combined using the operator “OR”. The combination of free vocabulary and/or medical subject headings (MeSH) terms for the identification of studies in PubMed, Scopus, and the Web of Science Core Collection were reported in Table 1.

#### 2.2.3. Selection Process 

After the articles were submitted to a public reference manager (Mendeley Desktop 1.19.8) to eliminate duplicates, possible relevant articles were screened by two reviewers using title and abstract (FS and DC). Studies that did not meet the inclusion criteria were excluded from the review and any disagreement was resolved through discussion until a consensus was reached. Subsequently, the remaining studies were included in the final stage of data extraction.

#### 2.2.4. Data Collection Process and Synthesis Methods

The data extraction and synthesis process started with cataloguing the studies’ details. To increase validity and avoid potentially omitting findings for the synthesis, the two authors (FS and DC) extracted information and completed a table taking into consideration: patient age, LMS treatment, symptoms at presentation, bone metastases diagnosis, time from primary diagnosis to bone metastasis, main bone spine lesion level, other bone metastases lesions or extra-osseous, the presence of a fracture, surgery type and approach, surgical complications, post-op chemo-radiotherapy, and time to last follow-up or death.

#### 2.2.5. Assessment of Methodological Quality

The methodological quality of the selected studies was independently assessed by two reviewers (DC and FS), using the quality assessment tools of the National Heart, Lung, and Blood Institute (NHLBI) [13]. In cases of disagreement, the reviewers attempted to reach consensus through discussion.

#### 2.2.6. Statistical Analysis 

Statistical analysis was performed using R software v.4.2.1; the R software packages ‘survival’ (v.3, 3-1), and ‘survminer’ (v.0.4.9) were used [14,15,16]. Data are reported at significance levels of *p* < 0.05. Survival analysis was performed to evaluate whether some variables of the initial conditions (e.g., recurrence, the presence of extra-osseous metastases) could influence the postoperative survival and clinical outcome of patients affected by spinal bone metastasis of uterine LMS. Survival analysis was performed using Kaplan–Meier analysis with patient death as an endpoint. Cox regression analysis was used to identify variables that can affect survival and to define the hazard ratio (HR) in cases of treatment failure. Survival analysis was performed by adjusting the models for age.

## 3. Results

### 3.1. Summary of Cases

In this retrospective case series, from 2012 to 2022, nine patients (1.16%) with uterine LMS spinal bone metastases were retrieved. All of the patients were Caucasian, with a mean age of 57 years (95% CI [51.9, 62.1]). Four patients had others medical comorbidities, of those, three had hypertension and one had gastroesophageal reflux. In one patient, hypertension was associated with other comorbidities, i.e., osteoporosis, colon tubular adenoma and hypothyroidism, while in another patient, hypertension was associated to type 2 diabetes, chronic venous insufficiency, gastritis, and chronic fatty liver disease. None of the patients had a history smoking and/or alcohol. Except for one patient, who had not yet been treated for primary uterine LMS, all of the patients were undergoing the menopause, naturally or induced by a hysterectomy. Almost all of the patients (6/9) had received previous chemotherapy and/or radiation therapy for primary tumors (Table 2).

The time between diagnosis of a primary tumor to diagnosis of bone metastases ranged from 0 month to 14 years, with only one patient having a diagnosis of bone metastasis at the same time of primary tumor diagnosis (mean time 5.2 years, 95% CI [2.9, 7.5]). Bone metastases were diagnosed by at least two investigative tools, including computed tomography (CT), plain radiography (RX), magnetic resonance imaging (MRI) or, in one case, with positron emission tomography (PET). Bone metastases were prevalently located in the thoracic spine (n = 8) followed by metastases located in the lumbar spine (n = 3), and the cervical spine (n = 2).

Three patients had extra-osseous metastases detected along with bone metastases. Pathological fractures were present in two patients and spinal pain (back pain, radiating pain, or both) was present in all patients before spinal surgery, with three patients with concurrent paresthesia and one patient with functional impotence of the cervical spine. SINS score detected indeterminate (possibly impending) instability in sevem patients and complete instability in two patients. Surgical staging of bone metastasis, evaluated by the Weinstein–Boriani–Biagini classification, indicated that the 44.4% of bone metastases resided within the anterior region of the vertebral element (sectors 4–9), another 44.4% within the anterior and posterior area (within sectors 3–10, 2–7), and one bone metastatic lesion within the posterior vertebral area (within sectors 5–8). When these lesions were analyzed according to the tissue layers involved it was found that 77.7% of cases showed an extradural extra-osseous extension and paraspinous envelope (layers A, B, C, and D). In one patient, an extra-osseous extension into the paraspinous region only, i.e., layers A, B, and C, was detected while another patient had an extra-osseous extension into the spinal canal only, i.e., layers B, C, and D. Six patients had an E preoperative Frankel score (typical motor and sensory function, pathological reflexes possible), two patients had a D score (proper function present below the plane of injury, walking with crutches) and one had a C score (some muscles below the level of injury have motor function, but no proper function). The post-operative Frankel grade classification was of grade E in six patients and grade D in three patients. At the last follow-up, all of the patients had a Frankel classification of E, except for two patients who had a D and C classification. Four patients had a recurrence in the spine as evaluated by RX or CT. The mean post-operative follow-up was 20.8 (95%CI [9.1, 32.4]) months (Table 3).

Almost all of the patients showed moderate or extensive vascularity on preoperative selective angiography and were subjected to embolization 2–3 days before spinal surgery. As previously reported, pain symptoms due to instability or fractures, were present in all patients treated in our series and surgical treatment was performed for progressive pain or when either vertebral collapse or metastatic growth caused spinal cord compression. In one case, the surgery was palliative, while in the remaining cases, intralesional curettage was used. The aim of these techniques was the same: stabilization and decompression of neural structures and in 88.8% of cases, an instrumentation was also used. Except for one case, where an anterior approach was used, in all the other cases, a posterior approach was employed. In all of the patients, the plane of dissection had transgressed into the lesion (intralesional margin) with contamination of resected margins. A figure of 66.6% of patients underwent post-operative chemo- and/or radiotherapy. One patient experienced a late post-surgical complication, i.e., a breakage of the instrumentation (Table 4).

### 3.2. Systematic Review

#### 3.2.1. Study Selection and Characteristics

The initial literature search retrieved 39 studies. Of those, 9 studies were identified using PubMed, 19 were identified using Scopus, and 11 were found in the Web of Science Core Collection. The articles were submitted to a public reference manager to eliminate duplicates and then screened for their titles and abstracts. Ten articles were selected and reviewed to establish whether the publications met the inclusion criteria, and six studies were considered eligible for this review. One study was not found. The search strategy and study inclusion and exclusion criteria are detailed in Figure 1. Of these articles, two were a case series and four were a case report.

#### 3.2.2. Assessment of Methodological Quality

In our quality assessment, 67% of the studies were rated as strong, 11% were rated as moderate, and 22% were rated as weak. Methodological weaknesses that led to moderate or weak quality scores often included the absence of well-described statistical methods-, consecutive cases, and comparable subjects. The risks of bias assessments for each study are reported in Table 5.

#### 3.2.3. Studies Results

A total of 6 articles describing 10 cases of bone metastasis to the spine were found by our search strategy (Table 6). The mean age at bone metastasis presentation was 49.3 years (95%CI [42.3, 56.3]). All of the patients were treated for primary LMS by hysterectomy or by bilateral salpingo-oophorectomy and only three of them received CHT and/or RT after primary tumor surgery [17,18]. The time between the diagnosis of a primary tumor to the diagnosis of bone metastases was 4.9 (95%CI [2.5, 7.3]) years. Bone metastases were diagnosed by different investigative tools, including mostly CT, RX, and MRI, but also PET and bone scintigraphy. The most common location involved was the lumbar spine (60%), followed by the thoracic spine (20%); in the remaining 20% of cases, metastases were located between the thoracic and lumbar spine. Two patients had extra-osseous lung metastases that were detected along with bone metastases diagnosis [18,19]. Pathological fractures were present in two patients [19,20] and back pain was the main reported symptom in all patients before spinal surgery, with three patients also having lower-extremity weakness, numbness, and heaviness [20,21] and one patient had tetraplegia [22]. Three patients underwent a total en bloc spondylectomy (TES) [18,19], two patients underwent a corpectomy and instrumentation [17,21], and five patients underwent a decompressive laminectomy and instrumented fusion [20,21,22], all through an anterior or posterior approach. A figure of 50% of patients underwent post-operative CHT and/or RT. Two patients developed post-operative tumor recurrence [21,22]. Of the 10 examined patients only 1 patient had a post-surgical complication, i.e., a deep-seated infection, and required instrumentation removal 6 months after and later fusion [21]. The mean post-operative follow-up was 56.1 (95%CI [34.1, 78.2]) months.

#### 3.2.4. Patient Survival

Among our retrospective case series and those collected from the literature, survival data after surgery for bone metastases were known only for 7 out of 9 and 8 out of 10 cases, respectively. The survival analysis was conducted on 7 cases for our retrospective series and on 8 cases for the literature data, and finally by considering all 15 cases. Figure 2 shows the probability of survival calculated according to the Kaplan–Meier estimator relating to the series of retrospective cases to that of cases collected from the literature and to all data of patients with bone metastases by LMS. A significant difference was found between the survival curves between the two case series populations considered (log rank test, χ^2^ = 3.7, *p* = 0.05). Overall, the median survival time for the retrospective case series was 46 months, while for the literature cases, it was 102 months. The overall median survival for all cases was 97 months. Cox regression analysis did not show the influence of variables such as the presence or absence of extra-osseous metastases or fractures, or post-operative treatment with chemo- and/or radiotherapy performed on the risk of developing bone metastases by LMS.

## 4. Discussion

Uterine LMS is an uncommon malignancy of the female genital tract with a poor 5-year survival rate [23]. It usually metastasizes early and to distant sites owing to a high propensity for hematogenous spread [24]. The lungs, peritoneum, and liver are relatively common sites of metastasis [24,25]. As shown by our systematic search, spinal bone metastases from uterine LMS are rare, with only two case series and four case reports published [17,18,19,20,21,22]. Almost all these spinal bone metastases from uterine LMS present destructive osteolytic lesions. As also demonstrated by our series, these features result in intractable pain, neurological deficits, and paraplegia [21]. Due to its rarity, disease management is based on these clinical cases, clinical status, and patient survival. To improve knowledge on spine metastases from uterine LMS, here, we analyzed a case series of patients who underwent excision of uterine LMS spinal bone metastasis and merged our series with surgically treated cases reported in the literature. The data from our cohort showed that spinal bone metastases from uterine LMS were present in 1.16% of patients who had proven uterine LMS over 10 years at a single institution. These patients and those from the literature always had bone metastasis in the spine only. Management of patients prior and after surgery often consisted of chemo- and/or radiotherapy. In these patients, spinal bone metastases from uterine LMS were more common at the thoracic and lumbar level. In all patients, surgery was safe and without major post-operative complications. As highlighted by our series, after surgery, all patients had stable or improved neurological outcomes, performance status, and pain as well as at the last follow-up. The cases in our series had a mean age of 57, which is higher than the mean age of 48 reported by the literature cases [17,18,19,20,21,22]. This is probably due to the recent progress made not only in the surgical approaches and therapy for advanced and recurrent primary disease, but also in the screening programs that offer a unique opportunity to detect signs of disease at early stages [26]. However, in our series this aspect does not seem to reflect on the overall survival. The overall median survival from cases derived from our systematic literature review was 102 months, while in our series, the overall median survival was 46 months. Thus, our series showed a shorter survival time after uterine LMS spinal bone metastasis. Considering the limited number of cases in our series, the difference in survival compared to the literature cases may be due to several factors, such as differences in grading, stage, baseline health status at presentation, presence of comorbidities, treatment of primary tumors, and different spinal bone metastasis surgical approaches. However, combining the information of our series with the literature series can support surgeons to better manage patients affected by these uncommon spinal lesions. In contrast to our series, where almost all patients received embolization prior to surgery, the literature series did not report data on preoperative embolization. Recently, preliminary data have reported that preoperative embolization of spinal metastatic tumors can enhance antitumor immune response [19]. However, the efficacy of embolization-stimulated antitumor immunological response was mostly reported in trans-catheter arterial embolization for hepatocellular carcinoma and renal embolization for renal cell carcinoma [27,28]. More generally, the role of preoperative embolization in decreasing intraoperative blood loss and its clinical significance are described in palliative surgery for spinal metastasis, which violates the tumor vessels in such a highly vascular condition [29,30]. However, the influence of preoperative embolization in reducing intraoperative blood loss in free margin excision is unclear. At our institution, spinal embolization is regularly performed 2–3 days before spinal surgery, to reduce the risk of unexpected intraoperative bleeding due to the injury of segmental arteries and veins and to prepare for unexpected intralesional tumor resection. We recognize that this study has some limitations, including the small cohort size and the retrospective nature of the analysis without controls, which potentially introduced imperfect validity of the analyses and bias. Furthermore, we could not obtain detailed information about the primary lesion (tumor size, pathologic stage, and classification) and systemic therapy (indication and duration) because the primary lesions were treated and therapies were performed at other hospitals (patients were referred to our institute for the purpose of spinal metastases). Prospective multicenter studies analyzing the primary lesion, functional status, and systemic therapy are necessary to greatly improve our understanding of the clinical outcomes and survival in this pathological condition. Despite these limitations, this study collected comprehensive data over a 10-year period and described a spectrum of clinical courses in patients treated surgically for spine metastasis from uterine LMS. For such patients, it was shown that surgery can help to maintain a better performance status and with a low rate of complications. Furthermore, this study, for the first time, examined a series of patients with a specific gynecological primary tumor, uterine LMS, that metastasized to a specific bone site, the spine. Our findings on the outcomes of surgical procedures can provide key information for the treatment of these patients since the wide heterogeneity of primary cancer and of bone metastases requires personalized and targeted treatments by the type of primary tumor and the site of metastasis. The results of this study may help clinicians in metastatic workup of these cancers and may also provide information on potential therapeutic targets and surgical approaches that are able to improve survival in these patients.

## Figures and Tables

**Figure 1 diagnostics-13-00015-f001:**
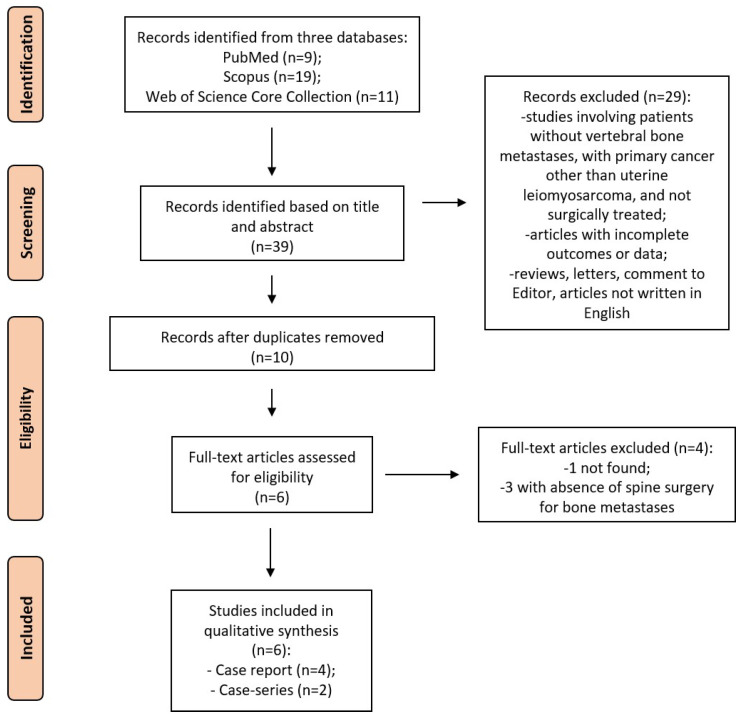
The PRISMA flow diagram for the systematic review detailing the database searches, the number of abstracts screened, and the full texts retrieved.

**Figure 2 diagnostics-13-00015-f002:**
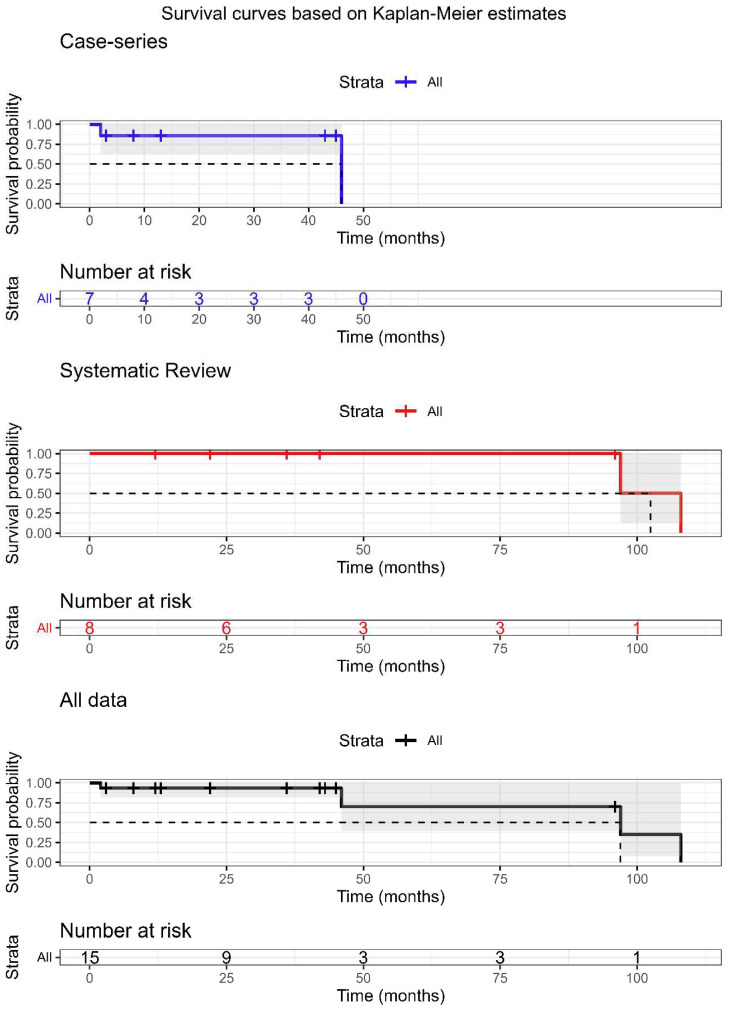
Survival analysis of patient with bone metastasis by LMS based on Kaplan–Meier estimates for the case series (*n* = 7), systematic review series (*n* = 8), and all data (*n* = 15). Cases with unknown follow-up outcomes or survival times were excluded from the analysis.

**Table 1 diagnostics-13-00015-t001:** Combination of free vocabulary and/or medical subject headings (MeSH) terms for the identification of studies in PubMed, Scopus, and the Web of Science Core Collection.

PubMed: (“spine” [MeSH Terms] OR “spine” [All Fields] OR “spines” [All Fields] OR “spine s” [All Fields] OR (“spinal” [All Fields] OR “spinalization” [All Fields] OR “spinalized” [All Fields] OR “spinally” [All Fields] OR “spinals” [All Fields]) OR (“spine” [MeSH Terms] OR “spine” [All Fields] OR “vertebral” [All Fields] OR “vertebrals” [All Fields])) AND ((“bone and bones” [MeSH Terms] OR (“bone” [All Fields] AND “bones” [All Fields]) OR “bone and bones” [All Fields] OR “bone” [All Fields]) AND (“metastasi” [All Fields] OR “neoplasm metastasis” [MeSH Terms] OR (“neoplasm” [All Fields] AND “metastasis” [All Fields]) OR “neoplasm metastasis” [All Fields] OR “metastasis” [All Fields])) AND ((“uterin” [All Fields] OR “uterines” [All Fields] OR “uterus” [MeSH Terms] OR “uterus” [All Fields] OR “uterine” [All Fields]) AND (“leiomyosarcoma” [MeSH Terms] OR “leiomyosarcoma” [All Fields] OR “leiomyosarcomas” [All Fields]))
Scopus: (TITLE-ABS-KEY (spine) OR TITLE-ABS-KEY (spinal) OR TITLE-ABS-KEY (vertebral) AND TITLE-ABS-KEY (bone AND metastasis) AND TITLE-ABS-KEY (uterine AND leiomyosarcoma))
Web of Science Core Collection: (TS = spine OR TS = spinal OR TS = vertebral) AND (TS = bone metastasis) AND (TS = uterine leiomyosarcoma)

**Table 2 diagnostics-13-00015-t002:** Patients’ individual general characteristics (n = 9).

Patient	Patient Presentation	Comorbidity	Smoking	Alcohol	Menopausal Status	LMS Treatment	Symptoms at Spinal Bone Metastasis Presentation
1	Caucasian/47-yo	None	No	No	Yes	Hysteroannessiectomy. Three cycles of CHT	Paraparesis and back pain
2	Caucasian/44-yo	None	No	No	Yes	Hysterectomy. CHT (ifosfamide and epirubicin)	Back pain
3	Caucasian/67-yo	Hypertension	No	No	Yes	Hysteroannessiectomy. Treatment NR	Back pain and functional impotence of the cervical spine
4	Caucasian/47-yo	None	No	No	No	Still to be treated	Palpatory pain in the cervico-thoracic passage and in the right paravertebral musculature that radiates to the ipsilateral scapula
5	Caucasian/58-yo	Osteoporosis, hypertension, tubular adenoma in colon, and hypothyroidism	No	No	Yes	Hysterectomy. No adjuvant therapy	Back pain
6	Caucasian/71-yo	Type 2 diabetes, hypertension, chronic venous insufficiency, gastritis, and chronic fatty liver disease	No	No	Yes	Hysterectomy. RT	Back pain
7	Caucasian/57-yo	None	No	No	Yes	Hysteroannessiectomy. CHT (adriamycin and dacarbazine)	Pain, paresthesia, and numbness in the lower limbs
8	Caucasian/62-yo	None	No	No	Yes	Hysteroannessiectomy. Four cycles of CHT	Back pain
9	Caucasian/60-yo	Gastroesophageal reflux	No	No	Yes	Hysterectomy. Treatment NR	Pain and right foot paresthesia

Abbreviations: yo: years old; CHT: chemotherapy; NR: not reported; RT: radiotherapy.

**Table 3 diagnostics-13-00015-t003:** Clinicopathologic characteristics of bone metastasis from LMS (n = 9).

Patient	Bone Metastases Diagnosis	Time from Primary Diagnosis to Bone Metastasis (Years)	Main Bone Spine Lesion Level	Other Bone Metastasis Lesions	Presence of Fracture	Extraosseous Metastases	Frankel Pre-op	Frankel Post-op	Weinstein–Boriani–Biagini Classification	Frankel at Latest FU	Time to Last FU or Death (Months)	Recurrence	Outcome
1	XR, CT, MR	5	T8	No	No	Yes	C	D3	4–9 A–D	D3	2	No	Deceased
2	XR, CT	6	T10	No	No	No	D1	D3	3–10 A–D	C	46	No	Deceased
3	XR, CT	7	C3-4	No	No	No	D2	D2	11–2 A–D	E	43	Yes	Alive
4	CT, MR	At the same time	T1-2	No	No	No	E	E	4–9 B–D	E	45	Yes	Alive
5	CT, XR	14	L4	No	No	Yes (lung)	E	E	4–8 A–D	E	13	No	Alive
6	CT, XR, MR	4	T5	No	No	Yes (lung)	E	E	2–7 A–D	E	3	Yes	Alive
7	CT	10 months	T3, L1	No	Yes	No	E	E	3–10 A–D	E	No follow-up	No follow-up	Unknown
8	PET-CT, MR	3	T12, L2	No	Yes	No	E	E	5–8 A–C	E	6	No	Unknown
9	CT, MR	7	T4	No	No	No	E	E	3–10 A–D	E	8	Yes	Alive

Abbreviations: XR: X-ray, CT: computed tomography; MR: magnetic resonance; PET: positron emission tomography; RT: radiotherapy; CHT: chemotherapy; FU: follow-up.

**Table 4 diagnostics-13-00015-t004:** LMS spine metastasis surgical approaches (n = 9).

Patient	Pre-op Chemo-Radiotherapy	Pre-Surgical Embolization	Surgery Type	Surgical Approach	Instrumentation	Surgery Margins	Contaminated Margins	Post-op Chemo-Radiotherapy	Surgical Complications
1	CHT, RT	Embolization of D7 and D8 vertebral pedicles	Curettage Laminectomy and vertebroplasty	Posterior	Yes	Intralesional	Yes	CHT	No
2	CHT, RT	Selective arterial embolization	Curettage Laminectomy	Posterior	Yes	Intralesional	Yes	RT	Yes (minor)
3	RT	No	Curettage Laminectomy and vertebroplasty	Posterior	Yes	Intralesional	Yes	None	No
4	None	Embolization of right and left intercostal artery of T5 and of the right intercostal artery of T7 up to the exclusion of flow	Curettage Laminectomy and vertebroplasty	Posterior	Yes	Intralesional	Yes	CHT, RT	No
5	CHT, RT	Embolization of L4 vertebral pedicles	Curettage Corpectomy	Anterior	No	Intralesional	Yes	None	No
6	CHT, RT	Embolization of the right supreme thoracic artery	Curettage Laminectomy and vertebroplasty	Posterior	Yes	Intralesional	Yes	RT	No
7	CHT, RT	No	Curettage Hemilaminectomy and arthrodesis	Posterior	Yes	Intralesional	Yes	None	No
8	CHT, RT	No	Palliative Arthrodesis	Posterior	Yes	Intralesional	Yes	RT	No
9	CHT	No	Curettage Vertebroplasty	Posterior	Yes	Intralesional	Yes	CHT, RT	No

Abbreviations: RT: radiotherapy; CHT: chemotherapy.

**Table 5 diagnostics-13-00015-t005:** Assessment of methodological quality.

Criteria	Cotangco et al., 2020	Lucas et al., 1996	Meltzer et al., 1992	Takemori et al., 1993 [17]	Kato et al., 2020 [18]	Yonezawa et al., 2019 [19]	Shirzadi et al., 2012 [20]	Elhammady et al., 2007 [21]	Nanassis et al., 1999 [22]
1. Was the study question or objective clearly stated?	**Y**	**Y**	**Y**	**Y**	**Y**	**Y**	**Y**	**Y**	**Y**
2. Was the study population clearly and fully described, including a case definition?	**Y**	**Y**	**Y**	**Y**	**Y**	**Y**	**Y**	**Y**	**Y**
3. Were the cases consecutive?	**N**	**N**	**N**	**N**	**Y**	**N**	**N**	**N**	**N**
4. Were the subjects comparable?	**NA**	**Y**	**Y**	**NA**	**Y**	**NA**	**NA**	**Y**	**NA**
5. Was the intervention clearly described?	**Y**	**Y**	**Y**	**Y**	**Y**	**Y**	**Y**	**Y**	**Y**
6. Were the outcome measures clearly defined, valid, reliable, and implemented consistently across all study participants?	**Y**	**Y**	**Y**	**Y**	**Y**	**Y**	**Y**	**Y**	**Y**
7. Was the length of follow-up adequate?	**Y**	**Y**	**Y**	**Y**	**Y**	**Y**	**Y**	**Y**	**Y**
8. Were the statistical methods well-described?	**NA**	**N**	**N**	**NA**	**Y**	**NA**	**NA**	**N**	**NA**
9. Were the results well-described?	**Y**	**Y**	**Y**	**Y**	**Y**	**Y**	**Y**	**Y**	**Y**

Abbreviations: Y: yes; N: no; NA: not applicable.

**Table 6 diagnostics-13-00015-t006:** General and clinicopathologic characteristics of previously published cases of bone metastasis from LMS.

Reference	Patient Presentation	LMS Treatment	Symptoms at Bone Metastasis Presentation	Bone Metastases Diagnosis	Time from Primary Diagnosis to Bone Metastasis (Years)	Main Bone Spine Lesion Level	Other Bone Metastases Lesions or Extraosseous Metastases	Presence of Fracture	Surgery Type and Approach	Surgical Complications	Post-op Chemo-Radiotherapy	Time to Last FU or Death (Months)
Takemori et al., 1993 [17]	47 yo	Hysterectomy and bilateral salpingo-oophorectomy RT (45 Gy)	Back pain	XR, MR, bone scintigraphy, ultrasonography	2	T8	No	No	Corpectomy with instrumentation. Anterior	No	CHT (cyclophosphamide, vincristine, adriamycin, dacarbazine)	NR
Kato et al., 2020 [18]	(1) 59 yo (2) 44 yo	Hysterectomy RT, CHT	Back pain, neurological symptoms of lower extremities	XR, CT, MR	(1) 3.6 (2) 2.1	(1) T12-L2 (2) L1	(1) No (2) Yes (lung)	No	(1) TES: anterior–posterior (2) TES: posterior	No	(1) No (2) RT (39 Gy), CHT	(1) 97 (dead) (2) 36 (alive)
Yonezawa et al., 2019 [19]	44 yo	Hysterectomy	Lower back pain	XR, CT, MR	2	L1	Yes (lung)	Yes	TES. Anterior and posterior	No	No	12 (lung metastases)
Shirzadi et al., 2012 [20]	84 yo	Hysterectomy and bilateral salpingo-oophorectomy	Severe lower back pain, heaviness in the legs	XR, CT, MR	4	L5	No	Yes	Laminectomy, with instrumentation. Posterior	No	RT (45 + 9 Gy)	NR
Elhammady et al., 2007 [21]	(1) 45 yo (2) 42 yo (3) 46 yo (4) 36 yo	(1) and (2) hysterectomy and bilateral salpingo-oophorectomy (3) and (4) hysterectomy	(1) Lower back pain (2) Lower back pain, weakness (3) Lower back pain, lower extremity numbness, (4) Lower back pain, menorrhagia	(1) MR (2) MR, PET (3) CT, MR, PET (4) CT, MR	(1) 0 (2) 12 (3) 14 (4) 6	(1) L2 (2) L3 (3) T11, L2 (4) L5	No	No	(1) L2 corpectomy and instrumentation. Posterior (2) and (4) Laminectomy with instrumentation. posterior (3) Decompression with instrumentation. Posterior	(1), (3) and (4) No (2) Deep-seated infection	(1) CHT (adriamycin, cisplatin), RT (2) and (3) No (4) CHT (adriamycin)	(1) 42 (alive) (2) 96 (alive) (3) 36 (alive) (4) 108 (dead)
Nanassis et al., 1999 [22]	46 yo	Hysterectomy	Back pain, tetraplegia	XR, MR, bone scintigraphy	3	T2-T3	No	No	Decompressive surgery	No	RT, CHT	22

Abbreviations: LMS: leiomyosarcoma; RT: radiotherapy; CHT: chemotherapy; yo: years old; XR: X-ray, CT: computed tomography; MR: magnetic resonance; PET: positron emission tomography; TES: total en bloc spondylectomy; NR: not reported; FU: follow-up.

## Data Availability

The datasets generated during and/or analyzed during the current study are available from the corresponding author on reasonable request.

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
