# Peer review of "Postoperative Survival and Clinical Outcomes for Uterine Leiomyosarcoma Spinal Bone Metastasis: A Case Series and Systematic Literature Review"

_diagnostics, 2022, doi:10.3390/diagnostics13010015_

Round 1
Reviewer 1 Report
The authors present a case-series and review on outcomes for uterine leiomyosarcoma metastases.
Overall, the manuscript should be spell-checked by a native speaker to facilitate the reading flow.
Introduction
The overview on the general topic is nicely written and followed-up by a brief summary of what to expect from this review.
Materials and Methods
Results
It should be further highlighted that just n=9 patients were reviewed.
Fig. 1: The sagittal spine image does not add to the figure substantially.
The Weinstein-Boriani-Biagini classification should be explained at its first appearance.
How stable were the spine metastases? Please add the SINS for each of the 9 patients.
Systematic review:
Why did you exclude due to "methodological weaknesses"? What were the main criteria?
What was the potential reason for you not having a survival rate comparable to the literature?
Discussion
-
Author Response
Overall, the manuscript should be spell-checked by a native speaker to facilitate the reading flow.
The manuscript was reviewed and checked by a native speaker.
Introduction
The overview on the general topic is nicely written and followed-up by a brief summary of what to expect from this review.
Materials and Methods
Results
It should be further highlighted that just n=9 patients were reviewed.
As suggested by the reviewer we highlighted that just n=9 patients were reviewed (Lines: 74, 165-166, 188, 213, 228).
-Fig. 1: The sagittal spine image does not add to the figure substantially.
As reported by the reviewer, considering that the Figure 1 does not add further information to our manuscript we eliminated the figure.
-The Weinstein-Boriani-Biagini classification should be explained at its first appearance.
As requested by the reviewer we explained the Weinstein-Boriani-Biagini classification at its first appearance (Lines: 91-97).
-How stable were the spine metastases? Please add the SINS for each of the 9 patients.
As suggested, we performed the SINS score for all patients and added in the manuscript the instability risk (Lines: 99-106; 194-195).
-Systematic review:
-Why did you exclude due to "methodological weaknesses"? What were the main criteria?
We did not exclude articles based on the "methodological weaknesses”, but we assessed the methodological quality through the Quality Assessment Tools of the National Heart, Lung, and Blood Institute (NHLBI) (reported in Table 5) evaluating methodological weaknesses for each of the reviewed articles.
-What was the potential reason for you not having a survival rate comparable to the literature?
As reported in the discussion section, considering the limited number of cases in our series, the different in survival compared to the literature cases may be due to several factors, such as difference in the grading, in stage, baseline health status at presentation, presence of comorbidities, treatment of primary tumors, different surgical approaches (Lines 324-328).
Reviewer 2 Report
This paper reports on the rare condition of spinal metastasis of uterine leiomyosarcoma, summarizing cases treated at the authors' institution and adding a literature review. Compared to sporadic reports in the past, this paper is of high clinical significance.
Although there is no need to comment on the content, one point is that the tables are difficult to read for the reader, which is very unfortunate. Please consider modifying the format of the tables, especially table 6.
Author Response
This paper reports on the rare condition of spinal metastasis of uterine leiomyosarcoma, summarizing cases treated at the authors' institution and adding a literature review. Compared to sporadic reports in the past, this paper is of high clinical significance.
Although there is no need to comment on the content, one point is that the tables are difficult to read for the reader, which is very unfortunate. Please consider modifying the format of the tables, especially table 6.
As suggested by the reviewer we modified the format of the Tables.